# Towards Unbiased Evaluation of Ionization Performance in LC-HRMS Metabolomics Method Development

**DOI:** 10.3390/metabo12050426

**Published:** 2022-05-10

**Authors:** Carsten Jaeger, Jan Lisec

**Affiliations:** Bundesanstalt für Materialforschung und -Prüfung (BAM), Department 1 Analytical Chemistry, Reference Materials, Richard-Willstätter-Straße 11, 12489 Berlin, Germany; jan.lisec@bam.de

**Keywords:** liquid chromatography-high resolution mass spectrometry, nontargeted analysis, method development, method harmonization, quality control, feature statistics, chemical classification, electrospray ionization

## Abstract

As metabolomics increasingly finds its way from basic science into applied and regulatory environments, analytical demands on nontargeted mass spectrometric detection methods continue to rise. In addition to improved chemical comprehensiveness, current developments aim at enhanced robustness and repeatability to allow long-term, inter-study, and meta-analyses. Comprehensive metabolomics relies on electrospray ionization (ESI) as the most versatile ionization technique, and recent liquid chromatography-high resolution mass spectrometry (LC-HRMS) instrumentation continues to overcome technical limitations that have hindered the adoption of ESI for applications in the past. Still, developing and standardizing nontargeted ESI methods and instrumental setups remains costly in terms of time and required chemicals, as large panels of metabolite standards are needed to reflect biochemical diversity. In this paper, we investigated in how far a nontargeted pilot experiment, consisting only of a few measurements of a test sample dilution series and comprehensive statistical analysis, can replace conventional targeted evaluation procedures. To examine this potential, two instrumental ESI ion source setups were compared, reflecting a common scenario in practical method development. Two types of feature evaluations were performed, (a) summary statistics solely involving feature intensity values, and (b) analyses additionally including chemical interpretation. Results were compared in detail to a targeted evaluation of a large metabolite standard panel. We reflect on the advantages and shortcomings of both strategies in the context of current harmonization initiatives in the metabolomics field.

## 1. Introduction

Metabolomics, a bioanalytical approach originally devised to study metabolic changes in biological model organisms, is now widely used in more applied fields of clinical diagnostics and pharmaceutical research [1,2]. Active adoption of metabolomics methodologies also takes place in areas such as toxicology, where they are expected to enhance regulatory frameworks [3]. With more “official” use and integration into routine analysis, method harmonization and standardization play more and more important roles [4,5], resulting in growing analytical demands and expectations toward nontargeted methods. It is commonly agreed that reproducibility and robustness in nontargeted analysis need to significantly improve, e.g., to allow long-term comparability in clinical diagnostics. Similarly, chemical comprehensiveness, in terms of sufficient analytical coverage of the chemical space, requires major enhancements to methods and instrumental setups [6]. To tackle these and related analytical challenges, suitable evaluation strategies are required to compare the benefits and drawbacks of methods and instrumental setups efficiently and systematically. Due to the complexity and high dimensionality of nontargeted data, however, such evaluation is not straightforward, and commonly accepted QC procedures are missing.

Bioanalysis in metabolomics makes use of two major technologies, nuclear magnetic resonance (NMR) and hyphenated mass spectrometry (GC-MS, LC-MS). NMR provides structure elucidation and quantitative capabilities for biomatrices in situ but requires relatively large sample amounts. GC-MS combines efficient chromatographic separation with sensitive detection but is limited to small volatile compounds or compounds made volatile by derivatization. LC-MS, by contrast, can detect larger metabolites natively due to the use of soft electrospray ionization (ESI), coupled directly to the LC effluent. Polar to medium polar molecules usually ionize well under ESI [7], with analyte response depending mainly on the chargeability and active surface area of the molecule and less so on molecular weight, resulting in a broad metabolite spectrum that can be analyzed.

Commercial LC-MS instruments and ESI ion sources, in particular, are continually refined to enhance operational robustness and sensitivity, addressing many of the above-named needs in nontargeted analysis. Technical enhancements apply to sprayer configuration and arrangement of ion optics (reviewed in [8]). Also, interfaces featuring redesigned gas sheaths and higher operating temperatures [9,10] have proven beneficial for improving the limits of detection for a range of compound classes (e.g., [11,12]). Potential drawbacks of applying additional heat and pressure to the ionization process include thermal degradation of labile analytes and amplification of background ions, resulting in no improvement of effective signal-to-noise ratios [13]. Apart from an earlier study [14], the impact of using different ion interfaces or changing LC-HRMS setups in related ways on nontargeted measurements has not been studied in detail. More importantly, evaluation strategies supporting the validation of analytical setups for long-term comparability of nontargeted results are lacking. This question, however, is expected to become more and more relevant with increased routine use of nontargeted analysis.

The goal of the present study was to investigate if a defined nontargeted test experiment can efficiently guide and support method development and evaluation of different instrumental setups. A minimal experiment was devised consisting of a few measurements of a test sample and extensive data analysis of the feature space, ensuring practicality and transferability. Targeted testing, comprising a large commercial metabolite library, was carried out for comparison, representing a common but time-consuming and costly procedure. Both strategies were exemplarily employed for evaluating two instrumental setups for their performance in nontargeted metabolomics: (1) LC-HRMS using a standard ESI interface as reference (REF); and (2) LC-HRMS equipped with a high-temperature “IonBooster” interface representing an alternative setup (ALT) with potentially improved performance. Results showed that similar conclusions about instrumental performance could be drawn from both strategies, suggesting that feature-based evaluation procedures can contribute to much-needed accelerated method development in nontargeted analysis.

## 2. Results

Selecting an optimal analytical method or instrumental setup for a given problem is a crucial step in analytical chemistry. Usually, the answer can be obtained empirically without much effort by generating and evaluating results from the different analytical options. In nontargeted analysis, however, evaluating such results in an unbiased manner is challenging. We were initially confronted with this problem when testing the ALT vs. the REF ion source setup for LC-HRMS-based urine metabolomics, aiming at enhanced ionization and reduced sample consumption. Identical sample amounts produced strikingly higher intensities in the ALT setup, and three-dimensional feature plots; “feature” referring to peaks defined by unique mass-to-charge [*m*/*z*] ratio and retention time [RT] pairs; looked significantly more populated (Figure 1A,B). Analyses were repeated using a different chromatography mode (reversed-phase chromatography, RPC) to ensure that observations were not singular effects of the hydrophilic liquid interaction chromatography (HILIC) method used first. Signal intensities in RPC were also consistently higher using the ALT setup (Appendix A), suggesting a clear sensitivity advantage independent of chromatography. However, despite the large offset in overall sensitivity, closer analysis of features revealed a significant subset of mass spectral features (8.6%) to be unique to the REF setup (Figure 1C). This indicated selectivity differences between both setups, which raised the question if overall metabolic profiles were ‘equivalent’ and could be compared, e.g., if setups were changed within long-term studies. This question prompted us to explore what information about selectivity could be derived from the feature space when analyzed in detail.

### 2.1. Evaluation of Selectivity

Differences in selectivity between analytical methods result in changes in metabolite intensities that, in nontargeted analysis, can be recognized from altered intensity distributions of the feature space. In the present case, the initial problem in conducting a fair statistical evaluation arose from the overall difference in sensitivity between the setups. Comparing samples at high concentrations resulted in visible overloading effects (signal saturation) in the more sensitive setup. Comparing samples at lower concentrations, by contrast, did not use the full ionization capacity of the less sensitive setup. We, therefore, investigated if a dilution series of the biosample could be used to avoid this problem. Eight sequential one-in-four dilutions of the original sample were prepared (1:1, 1:4, 1:16, …, 1:16,384) and submitted to LC-HRMS analysis. Feature intensities were then analyzed for ‘robust’ fold-changes considering all concentration levels, as outlined in Figure 2A. Results showed that feature intensities in the ALT setup were, on average, 4.3-fold higher (log_2_-fold change = 2.11) than in the REF setup when HILIC was used and 2.3-fold higher when RPC was used (Figure 2B,C). A total of 76–83% of the features had a higher response in the ALT setup, while for 17–24% of the features, the REF setup was more sensitive. As the feature space of nontargeted analyses can be compromised by both analytical and data analysis issues (see Section 3), we conducted a parallel targeted analysis of a metabolite standards library. Out of 604 metabolites, 303 and 322 exhibited acceptable peak shape using the HILIC and RPC methods, respectively. Determining peak intensities for the molecular ions ([M+H]^+^, [M−-H]^−^) of these compounds, we obtained positive fold-changes for the ALT setup similar to the nontargeted evaluation (Appendix A). While the absolute magnitude of changes was somewhat lower than in the nontargeted data, the same trend of higher average intensity gains using the HILIC method (2.4-fold) compared to using the RPC method (1.8-fold) was found for the ALT setup. This indicated, regarding instrumental setup, that the ALT setup performed better, that benefits were particularly high using the HILIC solvent system, and regarding evaluation based on nontargeted feature data, that conclusions drawn from this approach were well in line with targeted data.

### 2.2. Evaluation of In-Source Fragmentation

While the proposed calculations effectively reduce analytical variation, we realized that other factors such as in-source fragmentation could potentially vary between setups. Such issues are not considered in purely feature intensity-based statistics but can potentially distort results. For example, if a higher proportion of molecular ion species is affected by in-source fragmentation in one of the setups, more fragments appear as additional features in the feature table. This results in an overestimation of analyte number and thus method performance. We examined if such issues can also be detected based on nontargeted data, a step requiring features to be deconvoluted and combined into ‘compounds’. Various approaches exist for this purpose (e.g., [15]), mostly relying on the chromatographic correlation of related ions. Here, we chose a modified approach. We applied *findMAIN* [16] to identify *m/z* relationships of typical ESI ionization products ([M+H]^+^, [M+Na]^+^, [M+K]^+^ etc.) in the feature data. Components robustly supported by multiple adducts were then submitted to chromatographic deconvolution, adding correlated in-source fragments to the compound spectrum (see Section 4). Based on the compound spectra, sometimes also termed ‘reconstructed’ or ‘MS^1^’ spectra, the relative intensity of fragments of the total spectral intensity was determined as demonstrated in Figure 3A. This value allows for estimating the degree of in-source fragmentation. Evaluation for all spectra acquired on the REF setup showed that relative fragment intensity varied considerably between spectra, ranging between approx. 5 and 90% in most of the spectra (Figure 3B). This was likely due to the different nature and thermal stability of analytes. The average relative intensity was 40% and 55% for the HILIC and RPC methods, respectively. For the ALT setup, ranges, as well as averages, were similar with 46% and 49%, respectively, suggesting in-source fragmentation to be similar across setups. For comparison and to validate the approach, we also analyzed MS^2^ spectra acquired in the data-independent “broad-band CID” mode. Broad-band CID induces MS/MS-like fragmentation but does include precursor isolation so that spectra need to be deconvoluted exactly as MS^1^ spectra. As expected, such spectra exhibited much higher relative fragment intensities, averaging at 90% and 98% depending on chromatography (Figure 3C). We also examined spectra of metabolite standards for comparison to address uncertainties associated with nontargeted data analysis. Targeted evaluation of spectra, based on intensity ratio estimation as before (Figure 3A) but here including only confirmed adducts and fragments, revealed similar average values as in the nontargeted approach (30–42%; Figure 3D). The variation in relative fragment intensity across spectra was slightly smaller (12–72%). The absence of instrumental setup-specific differences in in-source fragmentation was clearly confirmed for the RPC method (40% vs. 42%), while for the HILIC method, the ALT setup differed somewhat from the REF setup (42% vs. 30%). This was possibly due to the different number of spectra available for statistical analysis for this combination of setup and chromatography (285 vs. 324). Still, nontargeted data seemed to provide good approximations of in-source fragmentation.

### 2.3. Evaluation of Ion Suppression

We next investigated if nontargeted data analysis also allowed conclusions on ion suppression, a frequent problem in ESI analysis of complex samples, and a problem potentially affecting instrumental setups to different extents [17]. Ion suppression is caused by different analytes or other components of the analytical matrix competing for ionization and can strongly reduce the linear dynamic range (LDR) attainably. This, in turn, hampers the identification of significant differences between biological samples. In nontargeted analysis, which lacks an absolute concentration scale, the true LDR in terms of analyte concentration range for which a linear signal response is observed is unknown and can at best be estimated on a relative scale. To attempt such an estimation, we analyzed the linear portion of the feature intensities obtained over the different dilution steps, technically by finding consecutive stretches of defined response ratios and fitting linear regressions through the longest stretch for each feature (see Section 4 for details). Another regression was fit to intensity values “above” (if any) and values within the first half of the linear stretch, assuming the presence of any ion suppression effects to cause a flattening in the response curve (Figure 4A). Applying this analysis to all 24,413 features of the HILIC subset resulted in valid LDR estimates (long enough stretches) in 11.3% and 27.7% of the cases for the REF and ALT setups, respectively. Summarizing slope changes by retention time, data suggested that the ALT setup was more affected by ion suppression than the REF setup in the feature-dense retention time range between 0 and 200 s (Figure 4B). Less clear differences were found for the later elution range (200–300 s). For validation, we compared these findings to a conventional assay of matrix effects using a post-column infusion of chemical standard compounds over a T piece and monitoring signal changes during a chromatography run. This assay similarly indicated higher signal suppression for the ALT setup in the early RT range (Figure 4C). Interestingly, most major suppression events, visible as downward spikes at approx. 25, 65, 95, 125, and 145 s (Figure 4C) were mirrored by similar events in the estimated data. Thus, estimation results appeared to represent ion suppression accurately. In addition, equivalent results for the RPC data were in line with expectations, showing the strongest suppression estimate near the column break-through as is typical for RPC and confirming more pronounced suppression in the ALT setup (Appendix A).

### 2.4. Evaluation of Chemical Comprehensiveness

We continued by exploring how potential differences in selectivity between instrumental setups could be analyzed in more detail and in a chemically more meaningful way. Feunang et al. (2016) [18] introduced ClassyFire, an algorithm generating chemical taxonomies for arbitrary chemical structures. The resulting chemical taxonomies consist of chemical classes at different hierarchical levels such as “kingdom”, “superclass”, and “subclass”. We hypothesized that categorizing all identifiable components within the feature space into such chemical class levels might reveal potential chemical bias in one setup compared to the other, information that would be immensely helpful during method development. Classifying features obviously requires prior feature identification, which is one of the biggest challenges in metabolomics. However, we reasoned that approximate identification might be sufficient for our purpose, as compounds with similar mass spectra would likely belong to the same chemical class. We tested the feasibility of this idea using the targeted dataset. Identifying and classifying spectra of 578 metabolites resulted in 83% correct classifications at the superclass level (e.g., “Organic acids and derivatives”), 78% correct classifications at the class level (e.g., “Carboxylic acids and derivatives”), and 71% at the subclass level (e.g., “Amino acids, peptides, and analogs”) (Appendix A). This relatively high percentage of correct assignments, especially at the higher ontological levels, encouraged us to similarly evaluate the nontargeted dataset. Here, the procedure consisted of identifying deconvoluted spectra (see Figure 3) with MS-FINDER [19] and applying ClassyFire to the results (Figure 5A). Between 333 and 541 spectra, depending on instrumental setup and LC method, were successfully processed. Feature intensities summed by chemical class indicated that using the REF setup, organic acids were the dominant compound class, followed by organoheterocyclic compounds, benzenoids, and lipids accounting for 21.7%, 12.1%, and 11.2% of total intensity, respectively. In line with expected chromatographic selectivity, polar organic acids were more abundant in HILIC, while lipids made up a higher percentage in RPC. Summed intensities of these major compound classes did not change substantially in the ALT setup, except for organic nitrogen compounds (10.3% vs. 3.7%) (Figure 5B). A more detailed investigation of the raw data revealed that mostly amines and quaternary ammonium salts contributed to this increase (Appendix A), suggesting that differences in chemical comprehensiveness, albeit small, can be detected based on nontargeted data.

## 3. Discussion

In metabolomics, sustained efforts are made to improve reproducibility and chemical coverage of metabolic screening methods. Strategies for ‘global metabolomics’ comprise improvements at all steps of the analytical workflow, including sample preparation, chromatography, ionization, MS and MS/MS technique, as well as data post-processing [20,21]. Evaluating alternative analytical methods or data analysis procedures against each other requires suitable test systems that identify all relevant benefits and drawbacks of each method. In the case of targeted analysis of a few metabolites, test approaches are relatively straightforward: a selection of target analytes is analyzed under identical analytical conditions as the biological sample, often employing isotope dilution techniques to eliminate matrix effects. For nontargeted methods, evaluation approaches are more complicated. One strategy is to add a set of internal standards representing major metabolite classes under investigation (e.g., amino acids, nucleotides, steroids) and to determine analytical recoveries for this standard set. This approach was used, for instance, to compare metabolite extraction efficiencies for different protocols applied to a given biological matrix (e.g., [22]). Some approaches do not add standards but compare the recovery of all identified endogenous metabolites, e.g., as obtained by GC-MS analysis [23]. Similarly, standardized reference materials (SRMs) such as NIST SRM for human plasma (SRM 1950) [24] were used to compare the detection of endogenous metabolites across laboratories [25]. SRM 1950 offers the advantage that concentrations of many endogenous metabolites are known; however, its use is relatively costly, and comparable SRMs for other matrices do not exist.

Restricting method evaluation to previously identified, biologically plausible compounds or to standards added to analysis avoids the problems connected to raw feature analysis (redundancy, technical artifacts) but also results in a failure to capture effects in unannotated biological signals. Thus, a significant percentage of *m/z* information remains unconsidered. The analyses performed here aimed at maximum unbiasedness and thus did not exclude any features except sporadic ones by working on robust median values of multiple concentration levels. That way, a maximum of acquired data is analyzed, resulting in a broader and more robust database for evaluation. While some method developers may favor a more targeted optimization of compound classes that are particularly relevant to the research questions, e.g., TCA cycle intermediates in cancer research, the majority of biomarker and discovery workflow developments should benefit from a ‘global’ strategy as presented here.

To test the strategy, we aimed to compare ESI ion source performance, a special case which, to the best of our knowledge, only a few authors have attempted so far. Pandher et al. (2012) [14] compared heated vs. non-heated ESI based on all human plasma features with coefficients of variation (CV) below 25% across technical replicates of biological origin. That approach aimed at excluding sporadic peaks, often resulting from chemical or electronic noise, to obtain more robust estimates of changes in feature numbers. Here, we expanded this approach in several ways. First, we included multiple concentration levels of the sample to not only improve robustness in feature number estimation but also to account for the large instrumental offset in sensitivity observed. Second, we tested if increased in-source fragmentation and ion suppression differed between the setups, two factors with a potentially strong effect on feature number and intensity. As a third difference, we categorized features into chemical classes, aiming at characterizing potential selectivity differences in a chemically more meaningful way. Our findings regarding ion suppression (Figure 4) and chemical bias (Figure 5) suggest that these questions should be considered in method comparisons involving ionization aspects. Other authors have addressed ESI performance in the context of the design of experiments aimed at systematic optimization of ion source settings [26,27] or in the context of developing strategies aimed at normalizing ionization efficiency [28]. These studies, however, were different in intention and methodology from the present work and will not be discussed further.

Nontargeted approaches can only be as accurate as the weakest link in the data analysis toolchain. For example, issues in peak detection, which still plague current preprocessing software [29], may lead to false-positive peaks and misinterpretation of the presence or absence of compounds. Chromatographic deconvolution algorithms often misinterpret in-source fragments as intact molecules, resulting in wrong annotations [30]. As a consequence of these and related issues, compound identification in current nontargeted workflows must always be considered putative [31]. These limitations directly affect the applicability of nontargeted approaches to method optimization, including the workflow evaluated here as an alternative to targeted method evaluation strategies. We were aware of the limitations in peak processing and compound identification and tried to attenuate the effect of analytical artifacts wherever possible, for example, by limiting the analysis to features reproducible in replicate measurements and by calculating intensity fold-changes robustly using multiple concentration levels. Still, the partially diverging findings on in-source fragmentation (increased in targeted, no change in nontargeted) probably reflect limitations in peak picking and chromatographic deconvolution. While in the targeted assay, the different ionization products and fragments of each target compound were precisely quantifiable, analysis in the nontargeted case had to rely on deconvoluted spectra. These mass spectra are obtained from MS^1^ ion traces, and in crowded chromatographic regions, deconvolution is known to become error prone. Further work is required to improve algorithms and to determine optimal noise and correlation thresholds, ideally in an automated manner. Recent progress in machine learning approaches, e.g., allowing “auto-deconvolution” of chromatography data [32], has the potential to make nontargeted data analysis considerably more robust and user-friendly, facilitating more widespread adoption.

Detecting chemical bias based on nontargeted analysis proved particularly challenging as it involves many preprocessing steps. At the same time, it is probably the analysis that provides the most valuable information to method developers. We showed previously that correct identification on a sum formula level can be obtained for approx. 75% of analytes in multiplex metabolite analysis resembling metabolite screening conditions [16]. Here, we obtained a similar rate of correct chemical classification (83% at the superclass level) for a large set of metabolite standards. The misclassification rate of at least 17% is not negligible; however, it is unlikely that it affects the evaluation of chemical bias/selectivity, as misclassifications are equally distributed over all features independent of the intensity of distribution. Nonetheless, alternative chemical classification strategies that do not depend on prior compound identification are potentially better suited for the purpose and will be explored in the future. For example, structural inference from MS^2^ fragments as performed by some annotation approaches [33,34] could prove advantageous.

Taken together, we tested in a small pilot experiment a nontargeted LC-HRMS method combined with feature-based statistics as an alternative to costly targeted evaluation. The approach successfully identified one of the tested instrumental setups as superior for the given analytical problem. Conclusions on selectivity, in-source fragmentation, ion suppression, and chemical bias were largely in line with complementary targeted evaluation, demonstrating that important analytical performance characteristics can be derived from the nontargeted feature space alone. We conclude that metrics based on the nontargeted feature space hold great potential and should be used more routinely, not only for method development but also for quality assurance and inter-laboratory comparisons. In regulatory environments, defined nontargeted test experiments could additionally involve the use of SRMs and ideally also reference datasets to boost robustness in nontargeted analysis. Efforts in the nontargeted community need to step up to promote the use of such materials and datasets and to make SRMs widely available for a broad range of biological matrices.

## 4. Materials and Methods

### 4.1. Preparation of Test Sample and Chemical Standards

As a biological test sample, the urine of a healthy volunteer was split into aliquots of 200 µL, transferred to microfuge tubes, centrifuged for 5 min at 21,000× *g*, and dried by vacuum rotation. Immediately before LC-MS analysis, samples were reconstituted in 200 µL 75:25 (*v*/*v*) acetonitrile/methanol for HILIC analysis or 20% methanol for RPC chromatography. Samples were spiked with 4 µL pharmaceutical mix 17 (Neochema, Bodenheim, Germany; composition see Appendix A) for quality control (QC) purposes. After another centrifugation step, samples were serially diluted in 8 steps in a ratio of 1:4 with the respective sample diluent. Aliquots of each dilution were transferred to screw-cap HPLC glass vials and placed into the autosampler. Pure sample diluent served as blank.

The commercially available MSMLS metabolite library containing 604 unique compounds (Sigma, Darmstadt, Germany) was analyzed as targeted control. Metabolite standards (5 µg) were reconstituted by adding 100 µL 5% methanol or 100 µL 3:3:1 (*v*/*v*/*v*) chloroform/methanol/water, respectively, to each well of 96-well plates according to the manufacturer’s instructions. A total of 15 to 24 metabolites were combined into 28 different master mixes, aliquoted, and dried down in an Alpha 2–4 vacuum rotator (Christ, Osterode, Germany). The concentration of each metabolite was 6.6 µg mL^−1^. For LC-MS analysis, two aliquots of each master mix were reconstituted in HILIC or RPC sample diluent, respectively, and injected into the LC-MS system.

### 4.2. LC-HRMS Analysis

A high-resolution quadrupole time-of-flight (QqTOF) mass spectrometer (Impact II; Bruker Daltonik GmbH, Bremen, Germany) was coupled to an ultra-performance liquid chromatography (UPLC) system (ACQUITY H-Class; Waters, Eschborn, Germany). As ion sources, the standard Apollo-II or the high-temperature “IonBooster” source were used, representing the REF and ALT setup, respectively.

Two chromatography methods were used. A hydrophilic liquid interaction chromatography (HILIC) method was established using a buffered solvent system (A: 50% acetonitrile + 10 mM NH_4_COOH, pH 3.0 [HCOOH]; B: 75% acetonitrile + 10 mM NH_4_COOH, pH 3.0 [HCOOH]) and a 75 mm × 2.1 mm × 1.7 µm BEH Amide column (Waters, Eschborn, Germany). The gradient was programmed as follows: 0 min 1% A, 0.3 min 1%, 5.3 min 65%, 5.31 min 99%, 5.7 min 99%, 5.71 min 1%, 7.5 min 1%. A flow rate of 0.8 mL min^−1^ was used, 0.5 mL min^−1^ of which were discarded post-column using a T piece to remain within the optimal working range of the standard ion source. The second method employed the reversed-phase (RPC) mode, using a 75 mm × 2.1 mm × 1.8 µm HSS-T3 column (Waters GmbH, Eschborn, Germany) and H_2_O/acetonitrile each with 0.1% formic acid as mobile phases. Gradient conditions were: 0 min 99% A, 0.3 min 99%, 5.5 min 50%, 5.6 min 1%, 6.5 min 1%, 6.6 min 99%, 7.5 min 99%. Injection volume was 5 µL for the biological sample, 0.2 µL and 1 µL for metabolite standards in positive mode, and 1 µL and 5 µL for metabolite standards in negative mode.

QqTOF analyses were carried out using a scan rate of 8 s^−1^, a scan range of 30–1000 *m*/*z*, a funnel 1 RF of 200 Vpp, a funnel 2 RF of 200 Vpp, a hexapole RF of 60 Vpp, an ion energy of 5 eV, a collison energy of 500–800 Vpp, a transfer time of 35–75 µs and a pre-pulse storage of 2 µs. Broad-band collision-induced dissociation (bbCID) mode used MS^E^-like alternating low (8 eV) and high (40–100 eV) collision energies. Ion source temperatures and gas flows were chosen according to the manufacturer’s recommendations for intermediate LC flow rates (0.3 mL min^−1^). For ESI(+) mode, these were as follows (IonBooster settings in brackets): capillary voltage 4500 V (1000 V), endplate offset −500 V (−400 V), nebulizer pressure 2.5 bar (4.1 bar), dry heater 200 °C (200 °C), dry gas [N_2_] 8 L min^−1^. For ESI(-) mode, settings were identical except for capillary voltage (ESI: 3500 V, IB: 1000 V). Settings only applicable to IonBooster were charging voltage (300 V) and vaporizer temperature (350 °C).

Mass calibration was performed against an ion series produced by sodium formate, spiked into the LC effluent at the end of each chromatographic run.

### 4.3. Matrix Effect Assay

An intermediate dilution of the biological sample (level 3, 1:16) without addition of internal standards was analyzed using the above HILIC method. Pharmaceutical mix 17, diluted 1:200 in acetonitrile/methanol 3:1, was infused post-column using a syringe pump and T piece. The infusion flow rate was 60 µL min^−1^, while the LC flow rate was 0.3 mL min^−1^ as described. Changes in signal intensities were monitored based on extracted base peak chromatograms (BPC) of the 17 target masses ([M+H]^+^ ± 1 mDa). As a blank reference, pure mobile phase A was used.

### 4.4. Feature Analysis

#### 4.4.1. Peak Detection

Raw Bruker (.d) files were calibrated in the vendor software (DataAnalysis) and exported to mzML files using msconvert (https://proteowizard.sourceforge.io/tools.shtml) accessed on 22 June 2021. Peak detection and alignment over samples were performed in R [35] using the xcms package [36]. Peak detection parameters in xcms were set as follows: method centWave, ppm 15, peakwidth (4, 12), prefilter (3, 300), noise 100.

#### 4.4.2. Peak Alignment

Features were aligned using xcms (method linear, bw 2). Alignments were verified by overlaid BPCs of QC compounds. For all subsequent evaluations (Section 2.1, Section 2.2, Section 2.3 and Section 2.4), features were required to be present in all four replicate measurements of at least one concentration level of one instrumental setup (minfrag 1 with sample groups defined accordingly).

#### 4.4.3. Chromatographic Deconvolution

Ions of the same compound, including molecular peaks, isotopes, adducts, fragments, etc., were grouped into pseudospectra using an iterative procedure. First, peaks with similar retention times were combined into time groups using the hierarchical clustering function in R. In each time group, the combination of adducts explaining the greatest proportion of intensity was identified with findMAIN [16], and all other peaks chromatographically correlated to these adducts were assigned to the spectrum. The peaks were removed from the time group, and the procedure was repeated for the remaining peaks until no further combination of at least three adducts was found.

#### 4.4.4. Identification and Classification

Deconvoluted spectra (Section 4.4.3) were batched-processed in MS-FINDER [19] as described [16]. MS-FINDER results were read back into R, and SMILES codes of identified compounds were submitted to ClassyFire [18] using the provided Ruby API. Invalid results occasionally returned by ClassyFire (approx. 5% of the queries) were discarded. Class frequencies and associated signal intensities were summarized using standard R functionality.

#### 4.4.5. Linear Regression Analysis

An R function was implemented to analyze ion suppression effects based on feature intensities. Intensity values of a feature were arranged in order of increasing sample concentration and log2-transformed. These values were analyzed for consecutive stretches exhibiting defined positive log_2_-fold changes (here: 1.5–8). If one such stretch reached or exceeded a defined minimum length (here: 3), 2 linear regression models were fit to the intensity values using standard R functionality (function ‘lm’), the first one to the longest identified stretch (s_1_) and a second one to the stretch spanning the first intensity value (highest concentration) to the center intensity value of s_1_. Slopes were extracted from both models (a_1_, a_2_), and their difference (Δa) was calculated and returned by the function. In case no stretch reached or exceeded the minimum length, a “missing value” was returned, and the feature was not considered in the overall analysis. After applying the function to all features, Δa values were averaged over 5 s retention time windows and visualized as shown.

#### 4.4.6. Further Analyses

Venn diagrams were produced using the VennDiagram R package.

## Figures and Tables

**Figure 1 metabolites-12-00426-f001:**
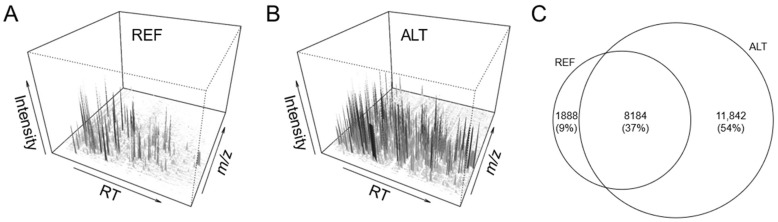
Demonstration of the initial problem. Two instrumental ion source setups (reference, REF; alternative, ALT) were tested for nontargeted analysis of an identical biosample. Results indicated large differences in overall sensitivity and feature landscapes (**A**,**B**). Despite lower sensitivity, a significant subset of features (8.6%) was exclusively detected using the REF setup (**C**).

**Figure 2 metabolites-12-00426-f002:**
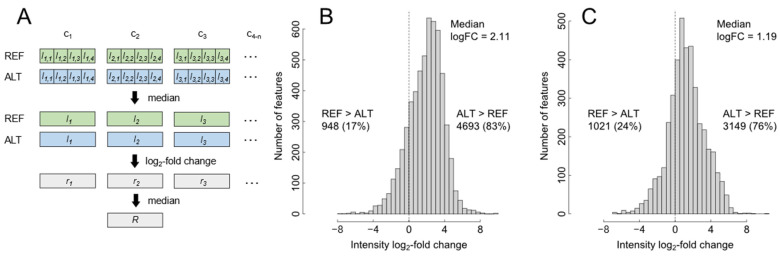
Evaluation of selectivity based on nontargeted data. Following analysis of a biosample by LC-HRMS at different concentrations levels, feature intensity values were extracted and expressed as log_2_-fold changes (logFC) between the two instrumental setups (ALT, REF), using the calculations outlined in (**A**): repeated measurements intensity values were median-summarized, expressed as log_2_-fold changes and again median-summarized. These ‘robust’ log-fold changes ALT vs. REF were visualized for the HILIC (**B**) and RPC (**C**) methods, respectively. Percentages of features with enhanced intensity in one of the setups are indicated beside the histograms. Abbreviations: c_1,n_, concentration level; i_j,m_, intensity value for concentration level (j) and technical replicate (m); r_j_, log2(i_j,ALT_/i_j,REF_) = logFC; R, robust logFC.

**Figure 3 metabolites-12-00426-f003:**
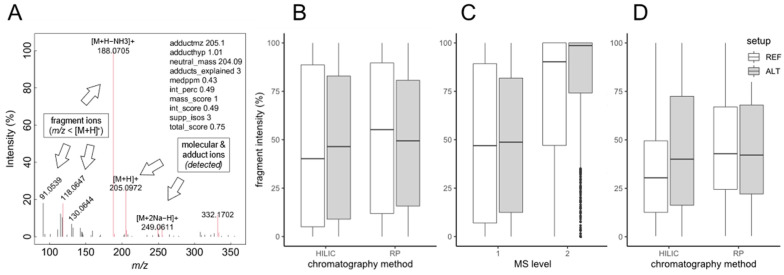
Evaluation of in-source fragmentation based on nontargeted data. (**A**) Exemplary MS^1^ spectrum (tryptophan, M = 204.0899) demonstrating the evaluation approach. In-source fragments were defined relative to the molecular ion that was inferred from adduct relationships. Summed fragment intensity was expressed relative to total intensity of annotated adducts (see arrows). (**B**) Summary of relative fragment intensity as a measure of in-source fragmentation across all spectra. Instrumental setup (REF, ALT) and chromatographic methods (HILIC, RPC) were compared as indicated. The number of analyzed spectra (from left to right) was *n* = 582/649/576/627. (**C**) Verification of the estimation approach by data-independent MS^2^ spectra, known to exhibit a higher degree of fragmentation due to the use of collision-induced dissociation (CID). Summary over both chromatography methods, *n* = 1158/1276/1121/1245. (**D**) Complementary targeted evaluation. LC-HRMS MS^1^ spectra of known metabolite standards were analyzed. Only fragments supported by chemically plausible sum formulas were considered. *n* = 285/324/324/340.

**Figure 4 metabolites-12-00426-f004:**
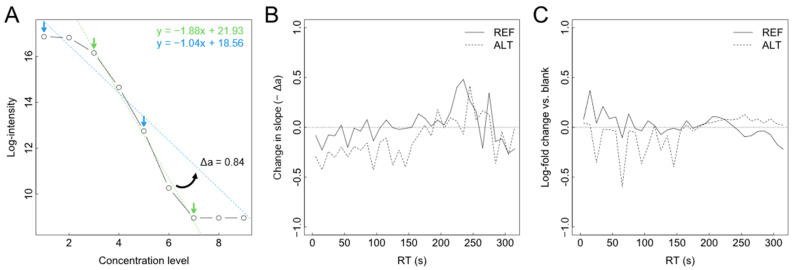
Evaluation of ion suppression based on nontargeted data. (**A**) A biosample was analyzed by LC-HRMS at different concentration levels, and each feature’s intensity series was submitted to a two-fold linear regression analysis. A first regression was fit through the linear stretch of the intensity series (green arrows), and a second regression was fit through the ‘flattened’ stretch at the beginning of the series (blue arrows). The change in slope (Δa) was determined as a potential marker of ion suppression. (**B**) Summary of Δa over the RT range for REF and ALT instrumental setups, respectively. Lines represent Δa values averaged over 5 s retention time (RT) windows. (**C**) Comparative assay of ion suppression using post-column infusion of chemical standards via a T piece. Summed intensities of 17 standard compounds expressed as log_10_-fold change vs. blank and averaged over 5 s RT windows.

**Figure 5 metabolites-12-00426-f005:**
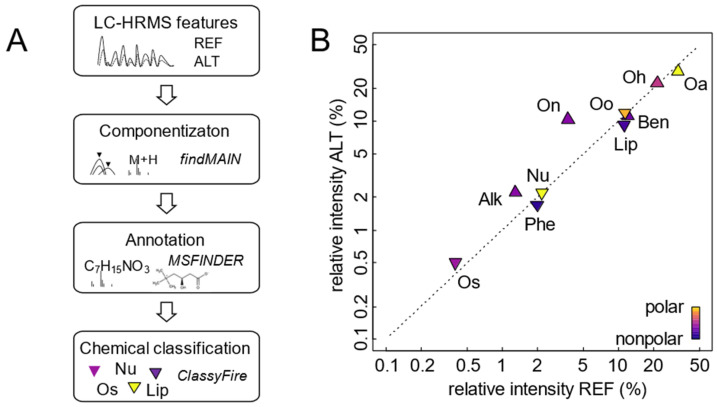
Evaluation of chemical comprehensiveness based on nontargeted data. (**A**) Workflow used summarize feature intensities by chemical class. (**B**) Comparison of relative abundances of chemical classes as detected with the two instrumental setups (REF, ALT). Symbols represent summed (log-) intensities of all features belonging to a class. Colors indicate median polarity (water-octanol partition coefficient; logP) of chemical classes, while symbol shape indicates higher total intensity in HILIC (△) or RPC (▽), respectively. Abbreviations correspond to ClassyFire “superclasses” (in order of decreasing intensity): Oa, organic acids and derivatives; Oh, organoheterocyclic compounds; Oo, organic oxygen compounds; Ben, benzenoids; Lip, lipids and lipid-like molecules; On, organic nitrogen compounds; Nu, nucleosides, nucleotides, and analogs; Alk, alkaloids and derivatives; Phe, phenylpropanoids and polyketides; Os, organosulfur compounds. Detailed results in Appendix A.

## Data Availability

The data presented in this study are available in the Appendix A.

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
