# Peer review of "Towards Unbiased Evaluation of Ionization Performance in LC-HRMS Metabolomics Method Development"

_metabolites, 2022, doi:10.3390/metabo12050426_

Round 1
Reviewer 1 Report
In this manuscript, the authors set up an unbiased procedure to evaluate the ionization performance for LC-HRMS based metabolomics method development. The authors applied the procedure to compare the performance of two instrumental ESI ion sources setups in terms of a number of features detected, selectivity, in-source fragmentation, ion suppression, and chemical comprehensiveness. The topic is interesting. I agree that the unbiased evaluation could be an alternative and rough comparison for the method development. However, this procedure is not a novel finding and the results in this manuscript are not very convincing.
The results need to be validated with targeted procedures due to the uncertainty of the untargeted findings. Moreover, the data processing steps for untargeted evaluation are much more comprehensive than the targeted ones. Although we can save time in selecting and preparing standards for the untargeted evaluation, much more time will be needed in the data processing.
Most of the time, method development is not a judgment of good or bad, it is a process of selection based on the needs. Different parameters will make your method focus on different metabolites, as the authors showed in the REF vs ALT methods. The selection will be based on the biological focus. The number of features is not enough to judge the method. The targeted metabolites data will give more information for the method development.
I agree that metrics should be used for method development. However, the metrics should be well characterized. It is another way of targeting compounds with metrics in them. Actually, lots of work and achievements have been made on this subject. SRM1950 and other standardized biological samples have been characterized for comparing among labs, and can also be used for method development. IROA tech has made yeast metabolite metrics with 5% and 95% 13C labeling pairs.
Reviewer 2 Report
The paper addresses the evaluation of non-targeted metabolomics methods using a pilot setup with two different ESI sources tested with both HILIC and reversed phase chromatography. A human urine sample and a sample consisting of a diverse comprehensive set of standards are used for testing. Authors evaluate chemical comprehensiveness and selectivity as well as the degree of in-source fragmentation and ion suppression across the tested methods.
The topic is highly relevant and there is certainly a need for testing and developing different approaches for evaluating non-targeted metabolomics analyses. The paper is well written, and I have only a the few, mostly minor, comments:
Line 91-92: Sentence needs minor revision: ”Identical sample amounts and produced strikingly higher intensities in the ALT setup”
Line 119: Sentence needs minor revision, there is an extra “the”: “…did not use the full the ionization capacity..”
Figure 2A: I suggest increasing the font size in the boxes, the text is difficult to read. In general, I find the font sizes used in figures 2 and 3 on the small side and would suggest increasing a bit to make it easier to read.
Line 218: The linear regression approach used to evaluate ion suppression is missing from the Materials and Method section. This information should be added before the paper is published.
Line 390 and 440: I suggest including details about the composition of the standard mix “Pharmaceutical mix 17”. Could be added in the supplementary.
Supplementary material: In the supplementary word file Figures S2 and S5 are too wide and therefore partly cut. The table is difficult to read because the columns are too narrow resulting in “stacked” numbers. I suggest re-sizing the two figures and the table and converting the word file to pdf. The table could also be included as a supplementary Excel file.
Reviewer 3 Report
The manuscript (metabolites-1646579) titled "Unbiased evaluation of ionization performance in LC-HRMS metabolomics method development" is very interesting. It is well-written fact that helps the readers. It is important for researchers that use general LC-MS in their research so I think it is suitable to be published.
Round 2
Reviewer 1 Report
Thanks for addressing my questions carefully.